# Rehabilitation status of children with cerebral palsy in Bangladesh: Findings from the Bangladesh Cerebral Palsy Register

Mahmudul Hassan Al Imam[1,2,3], Israt Jahan[1,2,3], Manik Chandra Das[1,2], Mohammad Muhit[1,2], Hayley Smithers-Sheedy[4], Sarah McIntyre[4], Nadia Badawi[4,5], Gulam Khandaker[1,2,6,7] *

1 CSF Global, Dhaka, Bangladesh, 2 Asian Institute of Disability and Development (AIDD), University of South Asia, Dhaka, Bangladesh, 3 School of Health, Medical and Applied Sciences, Central Queensland University, Rockhampton, Queensland, Australia, 4 Cerebral Palsy Alliance, Sydney Medical School, The University of Sydney, Camperdown, New South Wales, Australia, 5 Grace Centre for Newborn Intensive Care, Sydney Children's Hospital Network, Westmead, New South Wales, Australia, 6 Discipline of Child and Adolescent Health, Sydney Medical School, The University of Sydney, Sydney, New South Wales, Australia, 7 Central Queensland Public Health Unit, Central Queensland Hospital and Health Service, Rockhampton, Queensland, Australia

* gulam.khandaker@health.qld.gov.au

**Data Availability Statement:** The authors are unable to share the de-identified line listed data as the data contain potentially sensitive and identifying

## Abstract

### Objective

The objective of this study was to assess the rehabilitation status and factors associated with rehabilitation service utilisation among children with cerebral palsy (CP) in Bangladesh.

### Materials and methods

This is a population-based surveillance study conducted among children with CP registered in the Bangladesh CP Register (BCPR), the first population-based register of children with CP aged <18 years (y) in Bangladesh. Children with CP were identified from the community using the key informant method and underwent a detailed neurodevelopmental assessment. Socio-demographic, clinical and rehabilitation status were documented. Unadjusted and adjusted analyses with a 95% confidence interval (CI) were used to identify potential predictors of rehabilitation service uptake.

### Results

Between January 2015 and December 2019, 2852 children with CP were registered in the BCPR (mean (standard deviation, SD) age: 7 y 8 months (mo) (4 y 7 mo), 38.5% female). Of these, 50.2% had received rehabilitation services; physiotherapy was the most common type of service (90.0%). The mean (SD) age at commencement of rehabilitation services was 3 y 10 mo (3 y 1 mo). The odds of not receiving rehabilitation was significantly higher among female children (adjusted odds ratio (aOR) 1.3 [95% CI: 1.0–1.7], children whose mothers were illiterate and primary level completed (aOR 2.1 [95% CI: 1.4–3.1] and aOR 1.5 [95% CI: 1.1–2.1], respectively), fathers were illiterate (aOR 1.9 [95% CI: 1.3–2.8]), had

patient information; specifically sensitivities around the topic and due to the risk of participant identification given the specific/ defined study location and unique characteristics of participants. This is imposed by the Asian Institute of Disability and Development (AIDD) Human Research Ethics Committee (HREC) as part of the approval for the Bangladesh Medical Research Council ethics. Researchers may contact the AIDD for data access at the following: AIDD, House # 76 & 78, Road # 14, Block B, Banani R/A, Dhaka – 1213, Bangladesh; Phone: +88-02-55040839; Email: disabilityasia@gmail.com.

**Funding:** This study has been conducted as a part of the BCPR research project. The BCPR is funded by the Research Foundation of Cerebral Palsy Alliance (PG4314 – Bangladesh CP Register) and internal funding from CSF Global, Bangladesh. GK is supported by the Cerebral Palsy Alliance Research Foundation Career Development Fellowship (CDF 0116).

**Competing interests:** The authors declare that there is no conflict of interest.

a monthly family income ~US$ 59–118 (aOR: 1.8 [95% CI: 1.2–2.6]), had hearing impairment (aOR: 2.3 [95% CI: 1.5–3.5]) and motor severity (i.e. Gross Motor Function Classification System level III (aOR: 0.6 [95% CI: 0.3–0.9]) and level V (aOR: 0.4 [95% CI: 0.2–0.7])).

## Conclusions

Rehabilitation status was poor among the majority of the children with CP in the BCPR cohort, limiting their opportunities for functional improvement. A community-based rehabilitation model focusing on socio-demographic and clinical characteristics should be a public health priority in Bangladesh.

## Introduction

Childhood disability is a global public health concern due to its lifelong impact on physical and psychological wellbeing. An estimated 80% of childhood disability occurs in low- and middle-income countries (LMICs) [1]. Despite this high burden, there is limited information on access to rehabilitation services in LMICs [2]. The 2030 Agenda for Sustainable Development Goals warrants that children with disabilities should enjoy equal access to health care and rehabilitation (Goal #3 Good health and wellbeing) regardless of their abilities and socio-economic status (Goal #10 Reduce inequality).

Cerebral palsy (CP) is one of the leading causes of childhood disability, with an estimated 50 million people living with CP worldwide [3,4]. CP is a clinical description for non-progressive motor disorders caused by injury to the developing brain [5]. The burden of CP is estimated to be substantially higher in LMICs compared to high-income countries (HICs) [6,7].

Children with CP require support from a multidisciplinary team of medical and rehabilitation professionals including physiotherapists, occupational therapists and speech and language therapists to improve function, prevent secondary complications and enhance autonomy [8]. However, such services are not always available for children with CP, particularly in LMICs [9,10]. In Bangladesh, the population-based prevalence of CP was estimated to be 3.4 (95% CI 3.2–3.7) per 1000 children, which approximates to ~234,000 children with CP in a country of 166 million people [7]. However, this is likely to be an underestimation due to survival bias.

The utilisation of rehabilitation services for children with CP is multidimensional and is affected by many social, economic and ecological factors [11]. Younger age [12–19], male gender [20], high family income [7,10,16,21–24], parental education [7,22] and severe motor impairment [17,25–27] are positively associated with rehabilitation service uptake. Conversely, lack of access to information [10,24], lack of transportation support [7,22,24,25] and parents' lack of awareness [7,22,24,28] have been reported as barriers to rehabilitation service utilisation. However, a majority of these studies have been conducted in HICs [14,15,17–19,27,28]. Studies completed in LMICs to date have largely been conducted in hospital settings [13,20,22] and have focused on a particular service (i.e. physiotherapy) [22,24]. Furthermore, the limited service availability and shortage of rehabilitation services in LMICs make the situation more complicated.

Population-based data in this regard is limited in LMICs. Such data are essential to identify potential scope for intervention, ensure optimal use of inadequate available resources and maximise the service coverage for children with disabilities (e.g. CP) in the resource-constrained settings of LMICs like Bangladesh. Therefore, this study aimed to assess the

rehabilitation status and the factors associated with rehabilitation service uptake among children with CP in Bangladesh.

## Materials and methods

We established the first population-based surveillance of children with CP i.e. Bangladesh CP Register (BCPR) in rural Bangladesh in 2015. The BCPR is an ongoing surveillance that studies epidemiology, rehabilitation and intervention strategies to improve functional outcomes and limit associated impairments among children with CP in Bangladesh [29]. Initially, the surveillance activities (i.e. BCPR) were confined in one subdistrict (Shahjadpur, ~325 square kilometres, ~123,576 households) which represents rural and semi-urban Bangladesh (where the majority of the population (76.7%) lives [30]) in terms of demographic and other indicators (e.g. birth rate, immunisation rate, perinatal mortality rate, literacy rate). To date, this population-based surveillance has been maintained with high case ascertainment in Shahjadpur. Additionally, between 2015 and 2019 the BCPR has been scaled up to 17 other subdistricts (Σ4,338 square kilometres, ~1,304,960 households) following opportunistic recruitment at the community level (Fig 1).

### Study participants and data collection method

As part of the ongoing surveillance (i.e. BCPR), children with suspected CP in a community were identified using the key informant method (KIM). The KIM is a validated method where local volunteers (e.g. religious leaders, teachers, community health workers etc.) are trained as key informants (KIs) to identify children with disabilities in their communities [31]. The KIs were identified (approximately 1 KI per village) by the Community Mobilisers (CMs—paid project staff) and received a day-long training on the identification of children with suspected CP, disability sensitisation, advocacy for a disability inclusive society using flip charts, group work and role play. Following the training, the KIs were provided 4–6 weeks to identify and enlist children with suspected CP and share their contact details with CMs to bring those children and their primary caregivers to the nearest medical assessment camps for a confirmed diagnosis, detailed neurodevelopmental assessments and registration in the BCPR. The clinical definition of CP adopted from the Surveillance of CP in Europe (SCPE) [32] and the Australian CP Register (ACPR) [6] were strictly followed during case ascertainment. The details of case identification and recruitment into the BCPR have been described in our previous publication [7].

Data were collected using a standard case-record form (adapted from the ACPR) through interviews with the primary caregivers, as well as a clinical assessment and a review of medical records (if available). The detailed data collection method and the variables included in the BCPR are available in our previous publication [7]. In the current analyses, we used the following variables: (i) socio-demographic characteristics (e.g. age, gender, educational level of parents, monthly family income); (ii) clinical characteristics (e.g. Gross Motor Function Classification System (GMFCS) level, Manual Ability Classification System (MACS) level, predominant CP motor type, CP topography, associated impairments) and (iii) information on rehabilitation (i.e. whether the child ever received rehabilitation, the type and location/source of services received, age of commencement of rehabilitation). Here, the main outcome variable was the rehabilitation status (i.e. whether he/she ever received rehabilitation) of a child with CP registered in the BCPR, which is a binary variable with 'yes' and 'no' as responses. Information on rehabilitation was collected from primary caregivers of children with CP by a physiotherapist at the medical assessment camps in the context of BCPR recruitment. To document the type of rehabilitation services received, multiple responses were allowed and, therefore, the

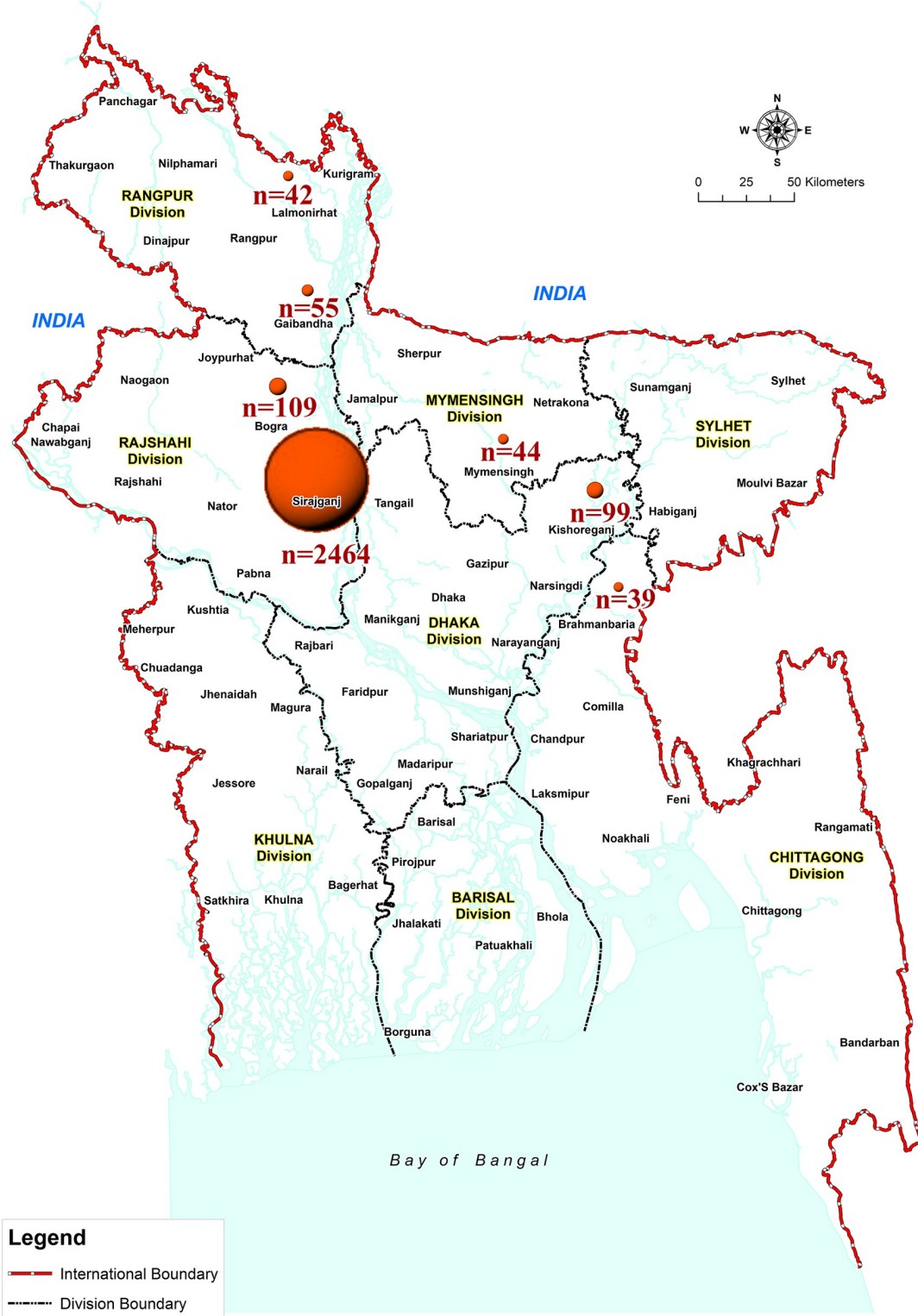

**Fig 1. Map of Bangladesh demonstrating the distribution of BCPR-registered children from each district in the study sample.** Circle diameters are proportional to the number of children with CP recruited in the study. (This map was produced by the authors using ArcGIS Desktop 10.8 software).

numbers presented for this variable are not mutually exclusive. Additionally, the medical records available from primary caregivers were reviewed for any documentation of rehabilitation services.

## Data management and analysis

Following collection, the data were entered electronically into the online password-protected BCPR data repository (http://bangladesh.cpregister.com/), access to which was limited to named investigators only. A dedicated data management team located at CSF Global in Dhaka, Bangladesh, with support from investigators conducted the data management and analyses.

To maintain the quality of the data, we used a double data entry method. Subsequently, data-entry error checks were performed by running frequencies of all variables to identify any outliers. In case of any missing data or incorrect/suspicious information, the BCPR case record forms (i.e. source documents) were reviewed. If the information was not found in the BCPR form, it was relayed to the field team and where possible the participants were contacted to gather the missing information. Continuous variables were collected as exact values and later recoded and categorised into groups (e.g. age was categorised as follows: 0–4 years, 5–9 years, 10–14 years and 15–18 years). Similarly, the monthly family income data were converted to United States Dollar (US$) (considering 1 US$ = 84.43 Bangladeshi taka (BDT)) and categorised into four family-income groups (i.e. BDT 500–4999 (US$ Σ6–59), BDT 5000–9999 (US$ Σ59–118), BDT 10,000–14,999 (US$ Σ118–178) and BDT 15,000 and above (US$ Σ178 and above). Comparison between the BCPR cohort and the general population was performed using the 2014 Bangladesh Demographic and Health Survey (BDHS) [33] and Household Income and Expenditure Survey [34] data from 2016. The poverty level among the families in the BCPR cohort was estimated using the national poverty lines (at the divisional level) as a cut-off. The proportion of families living below the poverty lines was then compared with the general population of the respective divisions as reported in the 2016 Household Income Expenditure Survey (HIES) in Bangladesh [34]. Bivariate analyses were completed to assess the impact of socio-demographic and clinical factors on rehabilitation status. For regression models, 'Not receiving rehabilitation' was considered as the main outcome of interest. Factors that were found to be statistically significant in unadjusted logistic regression were fitted into an adjusted logistic regression model. Odds ratios with 95% confidence intervals (CI) were reported. A $p$ value <0.05 was considered significant. All data were analysed using the Statistical Package for the Social Sciences (SPSS) software, version 26 (IBM, Armonk, NY, USA).

## Ethical considerations

Ethical approval for the BCPR study was obtained from the Cerebral Palsy Alliance Human Research Ethics Committee (EC00402; ref no. 2015-03-02) in Australia, the Asian Institute of Disability and Development Human Research Ethics Committee (southasia-irb-2014-l-01), and the Bangladesh Medical Research Council National Research Ethics Committee (BMRC/NREC/2013-2016/1267) in Bangladesh. Written informed consent was given by the parents or primary caregivers of all study participants prior to data collection and registration in the BCPR.

## Results

Between January 2015 and December 2019, 2852 children with CP were registered in the surveillance study (i.e. BCPR). The mean age at assessment was 7 years (y) and 8 months (mo)

(standard deviation (SD) 4 y and 7 mo; median 7 y 1 mo; interquartile range (IQR) 3 y 10 mo–11 y 4 mo); 38.5% (n = 1097/2852) female.

## Socio-demographic characteristics

The age and sex distributions of participating children were significantly different than the general population (children aged less than 10 y: 68.5% vs 49.0%; $p<0.001$, and male–female ratio 1.6:1 vs 1.1:1; $p<0.001$ in the BCPR vs the general population, respectively). Compared to the national data, fewer mothers and fathers of children in the BCPR cohort were educated (illiterate mothers: 30.0% vs 26.7%, and illiterate fathers: 39.2% vs 22.9%; $p<0.001$ in the BCPR vs the general population, respectively. Overall, 73.3% (n = 2074) children in the BCPR were from families living below the national poverty line, compared with 24.0% in the general population ($p<0.001$). The median monthly income of families was BDT 8000 (US$ ~95; IQR BDT 6000–10,000 (US$ ~71–118), mean BDT 9417 (US$ ~112) and SD BDT 7916.5 (US$ ~94)) (Table 1).

## Rehabilitation status

Almost half of the study participants (49.8%, n = 1411/2836) had never received any rehabilitation services. Among children who had received rehabilitation services, 90.0% (n = 1264/1404)

**Table 1. Socio-demographic characteristics of children with CP.**

| Characteristics | BCPR n (%) | General population % | p value |
|---|---|---|---|
| **Age group in years (n = 2799)[a]** | | | |
| 0–4 | 947 (33.8) | 23.5 | <0.001[e] |
| 5–9 | 972 (34.7) | 25.5 | |
| 10–14 | 621 (22.2) | 27.1 | |
| 15–18 | 259 (9.3) | 23.9[b] | |
| **Sex (n = 2852)** | | | |
| Male | 1755 (61.5) | 51.3[b] | <0.001[e] |
| Female | 1097 (38.5) | 48.7 | |
| **Mother's education (n = 2845)[a]** | | | |
| No education | 854 (30.0) | 26.7 | <0.001[e] |
| Primary | 1134 (39.9) | 35.3 | |
| Secondary and above | 857 (30.1) | 38.0[b] | |
| **Father's education (n = 2827)[a]** | | | |
| No education | 1108 (39.2) | 22.9 | <0.001[e] |
| Primary | 881 (31.2) | 37.6 | |
| Secondary and above | 838 (29.6) | 39.5[b] | |
| **Monthly family income, BDT (US$), (n = 2828)[a,c]** | | | |
| BDT 500–4999 (US$ Σ6–59) | 210 (7.4) | N/A | N/A |
| BDT 5000–9999 (US$ Σ59-–18) | 1704 (60.3) | N/A | N/A |
| BDT 10,000–14,999 (US$ Σ118–178) | 510 (18.0) | N/A | N/A |
| BDT 15,000 and above (US$ Σ178 and above | 404 (14.3) | N/A | N/A |
| **Families below the national poverty line (minimum–maximum according to divisions)** | 2074 (73.3) [22 (50.0) –1984 (74.2)] | 24.0 (16.0–32.8)[d] | <0.001[f] |

[a]Missing data exists (Age group in years = 53; Mothers' education = 7; Fathers' education = 25; Monthly family income = 24).

[b]Data from the 2014 Bangladesh Demographic and Health Survey (BDHS), and 23.9% for the general population refers to the 15–19 years age group [33].

[c]US$ 1 = BDT 84.43.

[d]Data from the 2016 Household Income and Expenditure Survey [34].

[e]Chi-squared test.

[f]Binomial test.

**Table 2. Rehabilitation status of children with CP in the BCPR cohort.**

| Variable Name | BCPR N = 2852 (%) |
|---|---|
| **Rehabilitation service received (n = 2836)[a]** | |
| No | 1411 (49.8) |
| Yes | 1425 (50.2) |
| **Type of rehabilitation service received (n = 1404)[a,b]** | |
| Physiotherapy | 1264 (90.0) |
| Advice | 156 (11.1) |
| Assistive device | 124 (8.8) |
| **Primary location of rehabilitation service (n = 1387)[a]** | |
| Non-governmental organisation centre | 626 (45.1) |
| Hospital | 443 (31.9) |
| Home-based | 153 (11.0) |
| Private clinic | 141 (10.2) |
| Special school | 24 (1.7) |
| **Age (y) at first commencement of rehabilitation service (n = 1365)[a]** | |
| <5 | 950 (69.6) |
| 5–10 | 324 (23.7) |
| 10 and above | 91 (6.7) |

[a]Missing data exists (Rehabilitation service received = 16; Type of rehabilitation service received = 21; Primary location of rehabilitation service = 38; Age at first commencement of rehabilitation = 60).
[b]Not mutually exclusive.

received physiotherapy. Of children with a GMFCS level III–V (73.7%, n = 2090/2836), only 4.7% (n = 98/2090) received any assistive devices. The mean age for commencing rehabilitation services was 3 y 10 mo (SD 3 y 1 mo; median 3 y 0 mo; IQR 1 y 6 mo–5 y 0 mo). Nearly one-third of children (30.4%, n = 415/1365) first received rehabilitation services at or over 5 y of age. The most common rehabilitation service providers were centres run by non-governmental organisations (45.1%, n = 626/1387) (Table 2).

## Factors influencing receipt of rehabilitation services among children with CP

**Age and sex of children with CP.** Children in the BCPR cohort aged 10–14 y and 15–18 y had 1.3 times (95% confidence interval (CI) 1.1–1.7) and 1.6 times (95% CI 1.2–2.2) lower likelihood of receiving rehabilitation services compared with children aged 0–4 y, respectively. Female children were 1.2 times (95% CI 1.0–1.4) less likely to receive rehabilitation services compared to male children (Table 3).

**Education of parents.** Parental education was significantly related to rehabilitation service uptake among children with CP in our cohort ($p<0.001$). Of the mothers and fathers of children with CP who had received rehabilitation services, 77.8% (n = 1105/1421) and 69.5% (n = 979/1409) had completed primary orsecondary education. Children of illiterate mothers had a 3.4 times (95% CI 2.8–4.2) lower chance of receiving rehabilitation services. A similar observation was reported for the father's education (Table 3).

**Monthly family income.** A significant negative association between monthly family income and rehabilitation service uptake was observed in our cohort. The median (IQR) monthly family income of children who had received and had never received rehabilitation services was BDT 8000 (6000–11,375) [US$ Σ95 (Σ71–135)] and BDT 7000 (6000–10,000) [US

**Table 3. Socio-demographic factors related to the receipt of rehabilitation for children with CP in the BCPR cohort.**

| Characteristics n = 2852 | Total n (%) | Ever received rehabilitation[a] | | | Unadjusted OR for not receiving rehabilitation (CI) | p value[d] |
|---|---|---|---|---|---|---|
| | | No n (%) | Yes n (%) | p value | | |
| **Age (n = 2799)[a]** | | | | | | |
| 0–4 | 947 (33.8) | 432 (46.0) | 507 (54.0) | 0.001[b] | Ref | |
| 5–9 | 972 (34.7) | 466 (48.1) | 502 (51.9) | | 1.1 (0.9 1.3) | 0.351 |
| 10–14 | 621 (22.2) | 331 (53.5) | 288 (46.5) | | 1.3 (1.1 1.7) | 0.004 |
| 15–18 | 259 (9.3) | 150 (58.4) | 107 (41.6) | | 1.6 (1.2 2.2) | <0.001 |
| **Sex (n = 2852)** | | | | | | |
| Male | 1755 (61.5) | 837 (47.9) | 910 (52.1) | 0.013[b] | Ref | |
| Female | 1097 (38.5) | 574 (52.7) | 515 (47.3) | | 1.2 (1.0 1.4) | 0.013 |
| **Mothers' education (n = 2845)[a]** | | | | | | |
| No education | 854 (30.0) | 534 (62.8) | 316 (37.2) | <0.001[b] | 3.4 (2.8 4.2) | <0.001 |
| Primary completed | 1134 (39.9) | 592 (52.6) | 533 (47.4) | | 2.3 (1.9 2.7) | <0.001 |
| Secondary and above | 857 (30.1) | 282 (33.0) | 572 (67.0) | | Ref | |
| **Fathers' education (n = 2827)[a]** | | | | | | |
| No education | 1108 (39.2) | 673 (61.0) | 430 (39.0) | <0.001[b] | 2.8 (2.3 3.3) | <0.001 |
| Primary completed | 881 (31.2) | 428 (48.9) | 448 (51.1) | | 1.7 (1.4 2.0) | <0.001 |
| Secondary and above | 838 (29.6) | 301 (36.2) | 531 (63.8) | | Ref | |
| **Monthly family income (n = 2828)[a]** | | | | | | |
| BDT 500–4999 (US\$ Σ6–59) | 210 (7.4) | 104 (50.2) | 103 (49.8) | <0.001[b] | 1.9 (1.4 2.7) | <0.001 |
| BDT 5000–9999 (US\$ Σ59–118) | 1704 (60.3) | 936 (55.3) | 758 (44.7) | | 2.4 (1.9 3.0) | <0.001 |
| BDT 10,000–14,999 (US\$ Σ118–178) | 510 (18.0) | 221 (43.6) | 286 (56.4) | | 1.5 (1.1 1.9) | 0.005 |
| BDT 15,000 and above (US\$ Σ178 and above | 404 (14.3) | 139 (34.4) | 265 (65.6) | | Ref | |
| **Median (IQR) monthly family income (n = 2812)[a]** | BDT 8000 (6000–10,000)/US\$ Σ95 (71–118) | BDT 7000 (6000–10,000)/US\$ Σ83 (71–118) | BDT 8000 (6000–11,375)/US\$ Σ95 (71–135) | <0.001[c] | | |

[a]Missing data exists (Ever received rehabilitation = 16; Age = 53; Mothers' education = 7; Fathers' education = 25; Monthly family income = 24).

[b]Chi-squared test.

[c]Mann–Whitney U test.

[d]Logistic regression.

$ Σ83 (Σ71–118)], respectively ($p<0.001$). Children from families with a monthly income of BDT 5000–9999 (US\$ Σ59–118) were 2.4 (95% CI 1.9–3.0) times less likely to receive rehabilitation services than children with a monthly family income of BDT 15,000 (US\$ Σ178) and above (Table 3).

**Predominant motor type and topography of CP.** Rehabilitation service utilisation was highest among children with dyskinesia and lowest among children with ataxia (52.3% (n = 92/176) vs 42.0% (n = 37/88), respectively). Compared to children with spastic CP, ataxic children were 1.4 times (95% CI 0.9–2.2) less likely to receive rehabilitation services. Furthermore, among children with spastic CP, tri/quadriplegic children had a 70% higher chance (95% CI 0.6–0.8) of receiving rehabilitation services than children with mono/hemiplegia (Table 4).

**GMFCS and MACS level.** GMFCS level III–V and MACS level III–V were significantly overrepresented among children who had received rehabilitation services (77.3%, n = 1092/1413, $p<0.001$, and 71.3%, n = 861/1207, $p = 0.001$, respectively). Children with GMFCS level

**Table 4. Clinical factors related to the receipt of rehabilitation for children with CP in the BCPR cohort.**

| Characteristics | Total n (%) | Ever received rehabilitation[a] | | | Unadjusted OR for not receiving rehabilitation (CI) | p value[e] |
|---|---|---|---|---|---|---|
| | | No n (%) | Yes n (%) | p value | | |
| **Predominant motor type of CP (n = 2852)** | | | | | | |
| Spastic | 2293 (80.4) | 1129 (49.4) | 1155 (50.6) | 0.401[c] | Ref | |
| Dyskinetic | 179 (6.3) | 84 (47.7) | 92 (52.3) | | 0.9 (0.7 1.3) | 0.663 |
| Ataxic | 91 (3.2) | 51 (58.0) | 37 (42.0) | | 1.4 (0.9 2.2) | 0.118 |
| Hypotonic | 289 (10.1) | 147 (51.0) | 141 (49.0) | | 1.1 (0.8 1.4) | 0.606 |
| **CP topography (n = 2293)** | | | | | | |
| Monoplegia and hemiplegia | 638 (27.8) | 350 (54.9) | 287 (45.1) | <0.001[c] | Ref | |
| Diplegia | 410 (17.9) | 219 (53.4) | 191 (46.6) | | 0.9 (0.7 1.2) | 0.628 |
| Triplegia and quadriplegia | 1245 (54.3) | 560 (45.3) | 677 (54.7) | | 0.7 (0.6 0.8) | <0.001 |
| **GMFCS level (n = 2836)[a]** | | | | | | |
| I | 252 (8.9) | 151 (60.2) | 100 (39.8) | <0.001[c] | Ref | |
| II | 494 (17.4) | 269 (54.9) | 221 (45.1) | | 0.8 (0.6 1.1) | 0.172 |
| III | 599 (21.1) | 298 (50.0) | 298 (50.0) | | 0.7 (0.5 0.9) | 0.007 |
| IV | 492 (17.3) | 258 (52.5) | 233 (47.5) | | 0.7 (0.5 1.0) | 0.049 |
| V | 999 (35.2) | 432 (43.5) | 561 (56.5) | | 0.5 (0.4 0.7) | <0.001 |
| **MACS level (n = 2220)[a,b]** | | | | | | |
| I | 305 (13.7) | 156 (51.3) | 148 (48.7) | 0.001[c] | Ref | |
| II | 391 (17.6) | 190 (49.0) | 198 (51.0) | | 0.9 (0.7 1.2) | 0.540 |
| III | 422 (19.0) | 193 (46.1) | 226 (53.9) | | 0.8 (0.6 1.1) | 0.163 |
| IV | 423 (19.1) | 199 (47.3) | 222 (52.7) | | 0.9 (0.6 1.1) | 0.282 |
| V | 679 (30.6) | 262 (38.8) | 413 (61.2) | | 0.6 (0.5 0.8) | <0.001 |
| **Type of associated impairment** | | | | | | |
| Epilepsy (n = 2835)[a,d] | 897 (31.6) | 428 (48.0) | 464 (52.0) | 0.197[c] | 0.9 (0.8 1.1) | 0.197 |
| Intellectual (n = 1944)[a,d] | 1074 (55.2) | 570 (53.4) | 497 (46.6) | 0.020[c] | 1.2 (1.0 1.5) | 0.020 |
| Visual (n = 2813)[a,d] | 462 (16.4) | 257 (56.1) | 201 (43.9) | 0.004[c] | 1.3 (1.1 1.6) | 0.004 |
| Hearing (n = 2835)[a,d] | 580 (20.5) | 364 (63.2) | 212 (36.8) | <0.001[c] | 2.0 (1.6 2.4) | <0.001 |
| Speech (n = 2834)[a,d] | 2132 (75.2) | 1054 (49.7) | 1066 (50.3) | 0.744[c] | 1.0 (0.8 1.2) | 0.744 |
| **Number of associated impairments (n = 1889)[a]** | | | | | | |
| None | 412 (21.8) | 212 (51.7) | 198 (48.3) | 0.001[c] | Ref | |
| 1–2 impairments | 870 (46.1) | 406 (46.9) | 460 (53.1) | | 0.8 (0.7 1.0) | 0.107 |
| 3–5 impairments | 607 (32.1) | 349 (57.9) | 254 (42.1) | | 1.3 (1.0 1.7) | 0.053 |

[a]Missing data exists (Ever received rehabilitation = 16; GMFCS level = 16; MACS level = 21; Epilepsy = 17; Intellectual impairment = 908; Visual impairment = 39;

Hearing impairment = 17; Speech impairment = 18; Number of associated impairments = 963).

[b]MACS was assessed among children aged four years of age or over.

[c]Chi-squared test.

[d]Reference category: No impairment.

[e]Logistic regression.

V were 50% more likely (95% CI 0.4–0.7) to receive rehabilitation services compared to children with GMFCS level I ($p<0.001$). A similar association was observed between rehabilitation service utilisation and the MACS level of children in the BCPR cohort (Table 4).

**Associated impairments.** Children with 3–5 associated impairments had a lower likelihood of receiving rehabilitation services than children with 1–2 impairments (odds ratio (OR): 1.3 (1.0–1.7) vs OR 0.8 (0.7–1.0), respectively). Furthermore, rehabilitation service utilisation was significantly lower among children with intellectual impairment (46.6%, n = 497/1067;

*p* = 0.020), visual impairment (43.9%, n = 201/458; *p* = 0.004) and hearing impairment (36.8%, n = 212/576; *p*<0.001) (Table 4).

**Independent predictors of not receiving rehabilitation services among children with CP in the BCPR cohort.** Child's gender, maternal and paternal education, monthly family income, GMFCS level, and the presence of hearing impairment were found to be significantly associated with rehabilitation service utilisation for children with CP registered in the BCPR when adjusted for other socio-demographic and clinical factors. The adjusted odds ratios (aORs) for not receiving rehabilitation were 1.3 (95% CI 1.0–1.7) for female children, 2.1 (95% CI 1.4–3.1) and 1.5 (95% CI 1.1–2.1) among children whose mothers were illiterate and primary completed, respectively, 1.9 (95% CI 1.3–2.8) among children whose fathers were illiterate, 1.8 (95% CI 1.2–2.6) among children with a monthly family income of BDT 5000–9999 (US$ 59–118), 0.6 (95% CI 0.3–0.9) and 0.4 (95% CI 0.2–0.7) among children with GMFCS level III and level V, respectively and 2.3 (95% CI 1.5–3.5) among children with hearing impairment (Table 5).

## Discussion

To the best of our knowledge, this is the first population-based study reporting the rehabilitation status and predictors of rehabilitation service uptake among children with CP in an LMIC. We observed that a large number of children with CP in the BCPR had never received any rehabilitation services. Children who had received services were more likely to be female and have educated parents, a higher socio-economic status and severe gross motor impairment.

Similar to this study, poor rehabilitation coverage has been reported among children with CP in India [22]. In contrast, Schmidt et al. [35] reported that 98.9% of children with CP in seven HICs had received rehabilitation services within one year. The proportion of children receiving rehabilitation services varies widely, even between LMICs. A higher proportion of rehabilitation service uptake has been observed in hospital-/institution-based studies, ranging from 55.6% in India [22] to 90.4% in Jordan [13], whereas studies conducted in community-based settings identified considerably poorer rehabilitation service uptake in Uganda (9.7%) [10] and South Africa (26.0%) [25]. The observed differences are most likely due to selection bias in hospital-/institution- and community-based settings and the socioeconomic conditions of study participants (i.e. more affluent people with a higher level of education are more likely to access services).

The age for commencing rehabilitation services among participants was substantially delayed when compared with HICs (3 y 8 mo in Bangladesh vs 1 y 5 mo in Australia) [36]. The reported delays in the diagnosis of CP in rural Bangladesh might play a key role here [7]. Recent evidence suggests that early initiation of rehabilitation services is crucial for the best possible motor outcomes [36].

Among children who had received rehabilitation services in the BCPR cohort, the majority received physiotherapy. Similar to our study, physiotherapy was frequently reported in studies conducted in Jordan (90.4%) [13] and Korea (81.3%) [12]. We found that more than two third of the children registered in the BCPR had MACS level III–V (68.6%) and could have benefitted from occupational therapy. Furthermore, 75.2% of children with speech impairment could have benefitted from speech therapy. However, our findings indicate that none of these children had received the required services. The low number of trained occupational therapists and speech and language therapists compared to physiotherapists (250 vs 260 vs 2400, respectively) with a higher availability in major cities in Bangladesh [37] might be responsible for this disparity. We also found that only 4.7% of children with a GMFCS level III–V had received

**Table 5. Predictors of not receiving rehabilitation service for children with CP in the BCPR cohort.**

| Characteristics | Not receiving rehabilitation[a] | |
|---|---|---|
| | Adjusted OR (CI) | *p* value |
| **Age** | | |
| 0–4 | Ref | |
| 5–9 | 0.9 (0.6 1.3) | 0.521 |
| 10–14 | 1.1 (0.8 1.6) | 0.546 |
| 15–18 | 1.4 (0.9 2.2) | 0.191 |
| **Sex** | | |
| Male | Ref | |
| Female | 1.3 (1.0 1.7) | 0.034 |
| **Mothers' education** | | |
| No education | 2.1 (1.4 3.1) | <0.001 |
| Primary completed | 1.5 (1.1 2.1) | 0.013 |
| Higher than primary | Ref | |
| **Fathers' education** | | |
| No education | 1.9 (1.3 2.8) | 0.001 |
| Primary completed | 1.2 (0.9 1.7) | 0.263 |
| Higher than primary | Ref | |
| **Family income** | | |
| BDT 500–4999 (US$ Σ6–59) | 1.6 (0.9 2.8) | 0.095 |
| BDT 5000–9999 (US$ Σ59–118) | 1.8 (1.2 2.6) | 0.004 |
| BDT 10,000–14,999 (US$ Σ118–178) | 1.2 (0.8 1.9) | 0.364 |
| BDT 15,000 and above (US$ Σ178 and above | Ref | |
| **CP topography** | | |
| Monoplegia and hemiplegia | Ref | |
| Diplegia | 1.3 (0.8 1.9) | 0.269 |
| Triplegia and quadriplegia | 0.9 (0.6 1.4) | 0.802 |
| **GMFCS level** | | |
| I | Ref | |
| II | 0.7 (0.5 1.2) | 0.167 |
| III | 0.6 (0.3 0.9) | 0.027 |
| IV | 0.7 (0.4 1.2) | 0.192 |
| V | 0.4 (0.2 0.7) | 0.002 |
| **MACS Level** | | |
| I | Ref | |
| II | 0.9 (0.6 1.4) | 0.644 |
| III | 0.9 (0.6 1.4) | 0.570 |
| IV | 0.7 (0.4 1.1) | 0.148 |
| V | 0.8 (0.5 1.4) | 0.453 |
| **Type of associated impairment[b]** | | |
| Intellectual | 1.3 (1.0 1.8) | 0.063 |
| Visual | 0.9 (0.6 1.4) | 0.662 |
| Hearing | 2.3 (1.5 3.5) | <0.001 |

[a]All variables found significant in the unadjusted analyses were included in the adjusted model to identify the potential predictors of not receiving rehabilitation services among children with CP in the BCPR.

[b]Reference category: No impairment.

assistive devices. The poor access to assistive devices found in this study is consistent with earlier studies conducted in LMICs [7,10,24]. Without mobility aids, children with severe functional motor limitations are likely to be bedridden and unable to participate in family, school and community life [10].

We also identified several socio-demographic and clinical factors as barriers in rehabilitation service uptake among our participating children. In terms of gender, female children with CP had a lower likelihood of receiving rehabilitation services when compared to male children. However, the current literature on this issue is conflicting. Young females with CP were found four times more likely to utilise rehabilitation services compared to males in the USA [38]. Whereas Sinha and Sharma [22] reported that there is no relationship between sex and rehabilitation service utilisation of children with CP in India. Women with disabilities face a double burden, because of their gender roles and disabilities, in LMICs [39]. Disability for a female becomes a greater barrier in terms of accessing opportunities such as rehabilitation [39,40]. McConachie et al. [20] described that having a male child can influence parents to seek rehabilitation services, particularly in rural settings. This might be related to the notion that male children need to be able to support a family in the future.

Parental education, and in particular maternal education, was significantly associated with rehabilitation service uptake among children with CP in the BCPR. Children whose parents were literate had significantly higher odds of receiving rehabilitation services. The findings are consistent with studies conducted in India [22] and the USA [38]. It is likely that parents who are educated are more aware of their child's health condition and needs, and understand the significance of providing rehabilitation to their children with CP. Additionally, parents with less education are more likely to be engaged in daily-basis low-paid jobs, which can make it difficult for them to take time away from work or to cover travel and/or service costs to get their children to rehabilitation centres. Our findings suggest that the majority of children in the BCPR were from impoverished families, and this cohort had higher odds of not receiving rehabilitation services. Financial constraints have been identified as a major barrier to utilising rehabilitation services in studies conducted both in HICs [38] and LMICs [10,22,24]. McConachie et al. [20] describe that a majority of children with CP cannot access rehabilitation services because of the costs, both direct (e.g. rehabilitation service charge) and indirect (e.g. transport cost, accommodation cost, food cost), associated with these services.

In terms of clinical factors, we found that children with GMFCS level III and V had a significantly higher likelihood of receiving rehabilitation services. This result is consistent with earlier studies conducted in Canada [17,26], the USA [26] and Australia [19]. The higher utilisation of rehabilitation services among severely motor-impaired children may be because of their increased rehabilitation needs in order to improve pain, comfort and quality of life. While the importance of rehabilitation for children with GMFCS level III–V cannot be undervalued, recent evidence suggests that the activity, function and participation of children with GMFCS level I–II could be enhanced through early intervention and rehabilitation services [41].

We also found that children with hearing impairment had a significantly lower probability of receiving rehabilitation services. Similar to our findings, Liljenquist et al. [38] found that children with CP and associated impairments had lower odds of utilising physiotherapy services in the USA. In contrast, Majnemer et al. [17] reported that children with lower intellectual impairment had higher odds of rehabilitation service uptake in Canada. It is not clear why children with hearing impairment have a lower likelihood of receiving rehabilitation in our cohort. Further study is required to investigate the effect of associated impairments on rehabilitation service uptake among children with CP in LMICs like Bangladesh.

## Strengths and limitations

This study used population-based data from an established register of children with CP in Bangladesh. Another methodological strength of this research is the adoption of the case definition of CP from the ACPR and the SCPE to ensure international consensus for the clinical diagnosis [6] and measurement of motor functions [42]. Despite our extensive efforts, this study had several limitations, however. Whilst the KIM is cost-effective in the identification of children with disabilities in LMICs [24,43], the BCPR recruitment efforts might have missed some children with CP who have mild motor limitations as the KIM has a 77.6% case-ascertainment rate compared with door-to-door surveying [31]. Therefore, children with severe motor limitations may have been overrepresented in this study. Secondly, the assessment of rehabilitation status was mostly based on the primary caregiver's responses, due to a lack of service utilisation records. Although there is a chance of recall bias, such a method has previously been used in HICs [17,19] and LMICs [10,24]. Thirdly, due to a lack of medical records, BCPR data collection has to rely on caregiver responses, in addition to the clinical examination, when assessing the severity of associated impairments in some of cases. Although this might have introduced information bias in determining the severity of associated impairments, such methods have also been previously used in other large-scale studies conducted in LMICs and HICs [7,44]. Finally, the national poverty lines were estimated based on household per capita consumption (food and non-food consumption/expenditure) using a detailed questionnaire from the HIES in Bangladesh [34]. Although information with that level of detail is not collected as part of the BCPR, the methodology is similar to some extent. However, there is still a risk of overreporting the poverty level with BCPR data due to the differences in survey tools and depth of information collected compared to the HIES, as well as several other factors.

## Conclusions

Nearly half of the children with CP in our study had not have access to rehabilitation services. A majority of children who were in need of assistive devices could not access them. Additionally, the age at commencement of rehabilitation was substantially delayed, limiting the opportunity to improve function and independence. Socio-demographic (i.e. sex, parental education and monthly family income) and clinical factors (i.e. GMFCS level and associated impairments) were significantly associated with rehabilitation service uptake. This evidence has important implications for policy formation and the improvement of rehabilitation services for children with CP in Bangladesh. Locally available and affordable early intervention and rehabilitation service delivery models, including training of rehabilitation professionals regarding community-based management of CP, should be seen as a priority for the strategic development of rehabilitation service coverage among this vulnerable population.

## Acknowledgments

We would like to express our heartfelt thanks to all primary caregivers and children with CP for their precious time and voluntary participation in the BCPR. We also would like to cordially thank the CSF Global team in Bangladesh for their diligent work and support in study implementation and guidance to primary caregivers to ensure necessary referral uptake.

## Author Contributions

**Conceptualization:** Mohammad Muhit, Gulam Khandaker.

**Data curation:** Mahmudul Hassan Al Imam, Israt Jahan, Manik Chandra Das, Gulam Khandaker.

**Formal analysis:** Mahmudul Hassan Al Imam, Israt Jahan, Gulam Khandaker.

**Funding acquisition:** Mohammad Muhit, Gulam Khandaker.

**Investigation:** Mohammad Muhit, Gulam Khandaker.

**Methodology:** Mohammad Muhit, Gulam Khandaker.

**Project administration:** Mahmudul Hassan Al Imam, Israt Jahan, Manik Chandra Das, Mohammad Muhit, Gulam Khandaker.

**Resources:** Mohammad Muhit, Gulam Khandaker.

**Software:** Gulam Khandaker.

**Supervision:** Mohammad Muhit, Hayley Smithers-Sheedy, Sarah McIntyre, Nadia Badawi, Gulam Khandaker.

**Validation:** Gulam Khandaker.

**Visualization:** Mahmudul Hassan Al Imam, Israt Jahan, Gulam Khandaker.

**Writing – original draft:** Mahmudul Hassan Al Imam.

**Writing – review & editing:** Israt Jahan, Manik Chandra Das, Mohammad Muhit, Hayley Smithers-Sheedy, Sarah McIntyre, Nadia Badawi, Gulam Khandaker.

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
