## [Decision Letter · Decision Letter 0]

10 Feb 2021

PONE-D-21-00848

Rehabilitation status of children with cerebral palsy in Bangladesh: findings from the Bangladesh Cerebral Palsy Register

PLOS ONE

Dear Dr. Khandaker,

Thank you for submitting your manuscript to PLOS ONE. After careful consideration, we feel that it has merit but does not fully meet PLOS ONE’s publication criteria as it currently stands. Therefore, we invite you to submit a revised version of the manuscript that addresses the points raised during the review process.

We look forward to receiving your revised manuscript.

Kind regards,

Enamul Kabir

Academic Editor

PLOS ONE

Journal Requirements:

2. Thank you for describing the provision of informed participant consent in the Methods section of your manuscript. We ask that you additionally provide this information in the Ethics Statement.

3.We note that you have indicated that data from this study are available upon request. PLOS only allows data to be available upon request if there are legal or ethical restrictions on sharing data publicly. For information on unacceptable data access restrictions, please see http://journals.plos.org/plosone/s/data-availability#loc-unacceptable-data-access-restrictions.

5.We note that Figure(s) 1 in your submission contain map images which may be copyrighted. All PLOS content is published under the Creative Commons Attribution License (CC BY 4.0), which means that the manuscript, images, and Supporting Information files will be freely available online, and any third party is permitted to access, download, copy, distribute, and use these materials in any way, even commercially, with proper attribution. For these reasons, we cannot publish previously copyrighted maps or satellite images created using proprietary data, such as Google software (Google Maps, Street View, and Earth). For more information, see our copyright guidelines: http://journals.plos.org/plosone/s/licenses-and-copyright.

a) You may seek permission from the original copyright holder of Figure(s) 1 to publish the content specifically under the CC BY 4.0 license. 

Additional Editor Comments:

The paper needs a major revision addressing the comments raised by two independent reviewers.

Reviewers' comments:

Reviewer's Responses to Questions

**Comments to the Author**

1. Is the manuscript technically sound, and do the data support the conclusions?

Reviewer #1: Yes

Reviewer #2: Yes

2. Has the statistical analysis been performed appropriately and rigorously? 

Reviewer #1: Yes

Reviewer #2: Yes

3. Have the authors made all data underlying the findings in their manuscript fully available?

Reviewer #1: No

Reviewer #2: No

4. Is the manuscript presented in an intelligible fashion and written in standard English?

Reviewer #1: Yes

Reviewer #2: Yes

5. Review Comments to the Author

Reviewer #1: Title: Rehabilitation status of children with cerebral palsy in Bangladesh: findings from the Bangladesh Cerebral Palsy Register

Objective: To assess the rehabilitation status and factors associated with rehabilitation service utilisation among children with cerebral palsy in Bangladesh.

Overall impression: A well conducted research. Introduction and justification of the study is good. Robust data analysis and rich discussion.

WHAT AUTHORS NEED TO IMPROVE ON

MAJOR

• Line 80 to 104, Line 309, abstract methodology; The paper has stated that data from the Bangladesh CP register was used. However, the methodology involved the researchers identifying the children in the community using key informat method. Data was collected from the children and caregivers and this data was recorded in the CP register.

This is not a study done on analysis of secondary data (Bangladesh CP register), or done from information gathered from secondary data(Bangladesh CP register). Rather, this study used primary data i.e. a community survey to identify children. Data gathered from the survey (from the children and care givers) was imputed into the BCPR.

Saying the study used data from Bangladesh CP Register (BCPR) is misleading (this portrays a study done by analyzing secondary data). This study correctly put is a POPULATION BASED STUDY DONE AMONG CP CHILDREN IN BANGLADESH. In addition to carrying out the study, data generated from the survey was imputed into the BCPR. This should be corrected in all aspects of the manuscript, including the abstract.

• Table 3 and Table 4. Please redo the bivariate analysis(chi square) using row total instead of column total. That way, the results will tally with the unadjusted odds ratio reported alongside.

• Table 3 and Table 4. Please indicate clearly on the COLUMN OF ‘UNADJUSTED OR’ if the outcome is ‘RECEIVED REHABILITATION’ or ‘NOT RECEIVED REHABILITATION’. This will make interpretation of the tables better and simpler,

Reviewer #2: Thank you for the paper. This is an important area of work in cerebral palsy research. However, there is need for some clarity on methods and results. See specific comments below.

Some additional point:

Introduction

• Very well written

Methods

• Provide some details of the Bangladesh CP Register (BCPR). How long has it been established for? How was surveillance area identified? Who collects data? Who is responsible for data management? Is data available publicly?

• Provide details of key informant method to enroll children into the registry.

• Page 6, line 104: Medical records reviewed were with primary caregivers or were they reviewed at the facilities where children received rehabilitation?

• Page 6, line 107: What data source was used for data on general population?

• What was the main outcome variable? Add details on how it is collected in BCPR.

• Provide details on how variables added to the analysis were handled.

• How was poverty level assessed?

Results

• Clarify “Illiterate mothers had 2.1 times (95% CI 1.8-2.5) less chance of receiving rehabilitation services” whether this relates to mothers receiving rehabilitation services of children of mothers with no education.

• Clarify “Children with a monthly income of BDT 5000-9999 (US$ ⁓59-118) were 1.7 [95% CI 1.5-2.0] times less likely to receive rehabilitation services than children with a monthly family income of BDT 15000 (US$ ⁓178) and above”. This is family or household income, not child’s income. Recheck.

Discussion

• Nicely written

General

• Mention full form at the first mention of an abbreviation

• Check format of the paper so that it is in accordance with the journal’s guidelines

• Check references, their formatting and citation style as per the journal’s requirements

• Check for language edits and sentence structure

Thank you.

6. PLOS authors have the option to publish the peer review history of their article (what does this mean?). If published, this will include your full peer review and any attached files.

Reviewer #1: No

Reviewer #2: **Yes: **Nukhba Zia

---

## [Author Response · Author response to Decision Letter 0]

13 Mar 2021

Professor Joerg Heber 

Editor-in-Chief

PLOS ONE

RE: Manuscript ID PONE-D-21-00848

Title: Rehabilitation status of children with cerebral palsy in Bangladesh: findings from the Bangladesh Cerebral Palsy Register

We would like to thank the academic editor and the reviewers for their constructive feedback and helpful comments on our manuscript titled ‘Rehabilitation status of children with cerebral palsy in Bangladesh: findings from the Bangladesh Cerebral Palsy Register’. Please see below our point-to-point responses to the academic editor’s and reviewers’ comments. 

Academic Editor:

Comment-1: Please ensure that your manuscript meets PLOS ONE's style requirements, including those for file naming. The PLOS ONE style templates can be found at

Our response: Thank you for the feedback. We have revised the title page and main body of the manuscript to meet the PLOS ONE style requirements. 

Comment-2: Thank you for describing the provision of informed participant consent in the Methods section of your manuscript. We ask that you additionally provide this information in the Ethics Statement.

Our response: Thank you for the appreciation and recommendation. We have now added the following statement in the Ethics Statement of the submission system.

‘Written informed consent was given by parents or primary caregivers of all study participants prior registration into the BCPR and data collection.’

Comment-3: We note that you have indicated that data from this study are available upon request. PLOS only allows data to be available upon request if there are legal or ethical restrictions on sharing data publicly. For information on unacceptable data access restrictions, please see http://journals.plos.org/plosone/s/data-availability#loc-unacceptable-data-access-restrictions.

Our response: Thank you for the suggestions. The authors are unable to share the de-identified line listed data as the data contain potentially sensitive and identifying patient information; specifically sensitivities around the topic and due to the risk of participant identification given the specific/ defined study location and unique characteristics of participants. This is imposed by the Asian Institute of Disability and Development (AIDD) Human Research Ethics Committee (HREC) as part of the approval for the Bangladesh Medical Research Council ethics. Researchers may contact the AIDD for data access at the following: AIDD, House # 76 & 78, Road # 14, Block B, Banani R/A, Dhaka – 1213, Bangladesh; Phone: +88-02-55040839; Email: disabilityasia@gmail.com.

Comment-4: PLOS requires an ORCID iD for the corresponding author in Editorial Manager on papers submitted after December 6th, 2016. Please ensure that you have an ORCID iD and that it is validated in Editorial Manager. To do this, go to ‘Update my Information’ (in the upper left-hand corner of the main menu), and click on the Fetch/Validate link next to the ORCID field. This will take you to the ORCID site and allow you to create a new iD or authenticate a pre-existing iD in Editorial Manager. Please see the following video for instructions on linking an ORCID iD to your Editorial Manager account: https://www.youtube.com/watch?v=_xcclfuvtxQ

Our response: Thank you for the suggestion. The ORCID iD of the corresponding author has been added.

Comment-5: We note that Figure(s) 1 in your submission contain map images which may be copyrighted. All PLOS content is published under the Creative Commons Attribution License (CC BY 4.0), which means that the manuscript, images, and Supporting Information files will be freely available online, and any third party is permitted to access, download, copy, distribute, and use these materials in any way, even commercially, with proper attribution. For these reasons, we cannot publish previously copyrighted maps or satellite images created using proprietary data, such as Google software (Google Maps, Street View, and Earth). For more information, see our copyright guidelines: http://journals.plos.org/plosone/s/licenses-and-copyright.

a) You may seek permission from the original copyright holder of Figure(s) 1 to publish the content specifically under the CC BY 4.0 license. 

Our response: Thank you for the detailed information and kind suggestion on this important issue. Fig 1 of the manuscript illustrates the study sites of the BCPR along with the number of children recruited into the BCPR from different districts of Bangladesh. We (authors) produced this map using ArcGIS Desktop 10.8 software. We assure you that there is no copyright issue associated with the map. Thus, this map can be made freely available online, and any third party is permitted to access, download, copy, distribute, and use the map in any way, even commercially, with proper attribution. This information has now been added into the methodology section of the revised manuscript. (Please see line 93 and 94 of the unmarked manuscript)

Comment-6: The paper needs a major revision addressing the comments raised by two independent reviewers.

Our response: Thank you for the feedback. We have revised the manuscript incorporating all comments of the reviewers. 

Reviewer: 1

Comment-1: 

Title: Rehabilitation status of children with cerebral palsy in Bangladesh: findings from the Bangladesh Cerebral Palsy Register

Objective: To assess the rehabilitation status and factors associated with rehabilitation service utilisation among children with cerebral palsy in Bangladesh.

Overall impression: A well conducted research. Introduction and justification of the study is good. Robust data analysis and rich discussion.

Our response: Thank you for your kind comments. 

Comment-2: Line 80 to 104, Line 309, abstract methodology; The paper has stated that data from the Bangladesh CP register was used. However, the methodology involved the researchers identifying the children in the community using key informat method. Data was collected from the children and caregivers and this data was recorded in the CP register.

This is not a study done on analysis of secondary data (Bangladesh CP register), or done from information gathered from secondary data(Bangladesh CP register). Rather, this study used primary data i.e. a community survey to identify children. Data gathered from the survey (from the children and care givers) was imputed into the BCPR.

Saying the study used data from Bangladesh CP Register (BCPR) is misleading (this portrays a study done by analyzing secondary data). This study correctly put is a POPULATION BASED STUDY DONE AMONG CP CHILDREN IN BANGLADESH. In addition to carrying out the study, data generated from the survey was imputed into the BCPR. This should be corrected in all aspects of the manuscript, including the abstract.

Our response: Thank you for the valuable suggestion. We have incorporated the suggestion and edited the manuscript accordingly. (Please see line 5-7, 79-82 of the unmarked manuscript) 

Comment-3: Table 3 and Table 4. Please redo the bivariate analysis(chi square) using row total instead of column total. That way, the results will tally with the unadjusted odds ratio reported alongside.

Our response: Thank you for the feedback. We have revised the Table 3 and Table 4 and added row percentages instead of column percentages. (Please see the Table 3 and Table 4 on page 12-14)

Comment-4: Table 3 and Table 4. Please indicate clearly on the COLUMN OF ‘UNADJUSTED OR’ if the outcome is ‘RECEIVED REHABILITATION’ or ‘NOT RECEIVED REHABILITATION’. This will make interpretation of the tables better and simpler.

Our response: Thank you for the suggestion. As recommended, we have inserted ‘Unadjusted odds ratios for not receiving rehabilitation’ in Table 3 and Table 4. (Please see the Table 3 and Table 4 on page 12-14)

Reviewer-2:

Comment-1: Thank you for the paper. This is an important area of work in cerebral palsy research. However, there is need for some clarity on methods and results. 

See specific comments below.

Our response: Thank you for your constructive feedback to improve our paper.

Comment-2: Introduction: Very well written

Our response: Thank you for the appreciation.

Comment-3: Methods: Provide some details of the Bangladesh CP Register (BCPR). How long has it been established for? 

Our response: Thank you for the valuable comment. We established the first population-based surveillance of children with CP i.e. Bangladesh CP Register (BCPR) in rural Bangladesh in 2015. The BCPR is an ongoing surveillance that studies epidemiology, rehabilitation, intervention strategies to improve functional outcomes and limit associated impairments among children with CP in Bangladesh [Khandaker et al. 2015]. Initially, the surveillance activities (i.e. BCPR) were confined in one subdistrict (Shahjadpur, ~325 square kilometres, ~123,576 households) which represents rural and semi-urban Bangladesh (where the majority of the population (76.7%) lives [Bangladesh Bureau of Statistics, 2014]) in terms of demographic and other indicators (e.g. birth rate, immunisation rate, perinatal mortality rate, literacy rate). To date, this population-based surveillance has been maintained with high case ascertainment in Shahjadpur. Additionally, between 2015 and 2019 the BCPR was scaled up to 17 other subdistricts (⁓4,338 square kilometres, ~1,304,960 households) following opportunistic recruitment at the community level (Fig 1). (Please see line 79-90 of the unmarked manuscript)

References: 

Khandaker G, Smithers-Sheedy H, Islam J, Alam M, Jung J, Novak I, et al. Bangladesh Cerebral Palsy Register (BCPR): a pilot study to develop a national cerebral palsy (CP) register with surveillance of children for CP. BMC Neurol. 2015;15:173. Epub 2015/09/27. doi: 10.1186/s12883-015-0427-9. PubMed PMID: 26407723; PubMed Central PMCID: PMCPMC4582618.

Bangladesh Bureau of Statistics. Bangladesh Population and Housing Census 2011. Dhaka, Bangladesh: Bangladesh Bureau of Statistics. 2014; [cited 2020 Aug]. Available from: http://203.112.218.65:8008/WebTestApplication/userfiles/Image/National%20Reports/Union%20Statistics.pdf

Comment-4: How was surveillance area identified? 

Our response: Thank you for the query. Our population-based surveillance area (i.e. Shahjadpur) was selected after assessing local demographic and other indicators (e.g. birth rate of Shahjadpur vs. Bangladesh, immunization of Shahjadpur vs. Bangladesh, perinatal mortality rate of Shahjadpur vs. Bangladesh, literacy of Shahjadpur vs. Bangladesh,). We wanted to select a sub-district that is most representative of rural and semi-urban areas of Bangladesh as 76.7% of areas are rural [Bangladesh Bureau of Statistics, 2014]. Moreover, we had existing community engagement and ongoing projects in the same community since 2003 which helped us to maintain a high case ascertainment rate in the surveillance area. All of these factors were considered while choosing the surveillance site for the BCPR. This information has now been added to the methodology section of the revised manuscript. (Please see line 82-88 of the unmarked manuscript)

Reference:

Bangladesh Bureau of Statistics. Bangladesh Population and Housing Census 2011. Dhaka, Bangladesh: Bangladesh Bureau of Statistics. 2014; [cited 2020 Aug]. Available from: http://203.112.218.65:8008/WebTestApplication/userfiles/Image/National%20Reports/Union%20Statistics.pdf

Comment-5: Who collects data? Who is responsible for data management?

Our response: Thank you for the valuable comment. As part of registration into the BCPR, data on selected variables were collected using a standard case record form adopted from the Australian CP Register by interviewing the primary caregivers, clinical assessment, reviewing medical records (if available). In the current analyses, we used the following variables: (i) socio-demographic characteristics (e.g. age, gender, educational level of parents, monthly family income); (ii) clinical characteristics (e.g. Gross Motor Function Classification System (GMFCS) level, Manual Ability Classification System (MACS) level, predominant CP motor type, CP topography and associated impairments) and (iii) information on rehabilitation (i.e. whether the child ever received rehabilitation, the type and location/source of services received, age of commencement of rehabilitation). 

Following data collection data were entered electronically into the password-protected BCPR online data repository (i.e. http://bangladesh.cpregister.com/) with access to named investigators only. A dedicated data management team located at CSF Global in Dhaka, Bangladesh with support from investigators conducts the data management and analyses. This information has now been added to the methodology section of the revised manuscript. (Please see line 111-128 of the unmarked manuscript)

Comment-6: Is data available publicly? 

Our response: Thank you for the query. The BCPR data is not available publicly. The BCPR data contain potentially sensitive and identifying information of children with CP and their primary caregivers. This information has been added to the Data Availability Statement.

Comment-7: Provide details of key informant method to enroll children into the registry.

Our response: Thank you for the valuable feedback. As part of the ongoing surveillance (i.e. BCPR), children with suspected CP in a community are identified using the key informant method (KIM). The KIM is a validated method where local volunteers (e.g. religious leaders, teachers, community health workers etc.) are trained as key informants (KIs) to identify children with disabilities in their communities [Mackey et al. 2012]. The KIs are identified (approximately 1 KI per village) by the Community Mobilisers (CMs—paid project staff) and receive a day-long training on the identification of children with suspected CP, disability sensitisation, advocacy for a disability inclusive society using flip charts, group work and role play. Following the training, the KIs are provided 4–6 weeks to identify and enlist children with suspected CP and share their contact details with CMs to bring those children and their primary caregivers to the nearest medical assessment camps for a confirmed diagnosis, detailed neurodevelopmental assessments and registration in the BCPR. The clinical definition adopted from the Surveillance of CP in Europe (SCPE) [Surveillance of cerebral palsy in Europe, 2000] and the Australian CP Register (ACPR) [Australian Cerebral Palsy Register Group, 2018] are strictly followed during case ascertainment. As suggested, we have now added the details with relevant references in our revised manuscript. (Please see line 96-110 of the unmarked manuscript)

References:

Australian Cerebral Palsy Register Group. Report of the Australian Cerebral Palsy Register, birth years 1995-2012; 2018. Australia; [cited 2020 Aug]. Available from: https://www.ausacpdm.org.au/resources/ australian-cerebral-palsy-register/

Mackey S, Murthy GV, Muhit MA, Islam JJ, Foster A. Validation of the key informant method to identify children with disabilities: methods and results from a pilot study in Bangladesh. J Trop Pediatr. 2012;58(4):269-74. Epub 2011/11/15. doi: 10.1093/tropej/fmr094. PubMed PMID: 22080830.

Surveillance of cerebral palsy in Europe. Surveillance of cerebral palsy in Europe: a collaboration of cerebral palsy surveys and registers. Surveillance of Cerebral Palsy in Europe (SCPE). Dev Med Child Neurol. 2000;42(12):816-24. Epub 2000/12/29. doi: 10.1017/s0012162200001511. PubMed PMID: 11132255.

Comment-8: Page 6, line 104: Medical records reviewed were with primary caregivers or were they reviewed at the facilities where children received rehabilitation?

Our response: Thank you for the query. The medical records available with primary caregivers were reviewed during medical assessment camps. The statement has been edited accordingly. (Please see line 126-128 of the unmarked manuscript)

Comment-9: Page 6, line 107: What data source was used for data on general population?

Our response: Thank you for the query. The data on age, sex and education of the general population were collected from the Bangladesh Demographic and Health Survey (BDHS), 2014 [National Institute of Population Research and Training (NIPORT), Mitra and Associates and ICF International, 2016]. The data on families below the international poverty line among the general population were collected from the Household Income and Expenditure Survey 2016 [Bangladesh Bureau of Statistics, 2019]. These data sources are mentioned in the Table 1 footnotes as well as in the methodology section of the revised manuscript. (Please see line 145-147, 186, 187 and 189 of the unmarked manuscript)

References:

National Institute of Population Research and Training (NIPORT), Mitra and Associates, ICF International. Bangladesh Demographic and Health Survey 2014. Dhaka, Bangladesh, and Rockville, Maryland, USA: NIPORT, Mitra and Associates, and ICF International: 2016. [cited 2020 Aug]. Available from: https://dhsprogram.com/pubs/pdf/FR311/FR311.pdf

Bangladesh Bureau of Statistics. Preliminary Report on Household Income and Expenditure Survey 2016. Dhaka, Bangladesh: Bangladesh Bureau of Statistics. 2019; [cited 2020 Aug]. Available from: http://bbs.portal.gov.bd/sites/default/files/files/bbs.portal.gov.bd/page/b343a8b4_956b_45ca_872f_4cf9b2f1a6e0/HIES%20Preliminary%20Report%202016.pdf

Comment-10: What was the main outcome variable? Add details on how it is collected in BCPR.

Our response: Thank you for the query. The main outcome variable was the rehabilitation status (i.e. whether he/she ever received rehabilitation) of a child with CP registered in the BCPR, which is a binary variable with ‘yes’ and ‘no’ as responses. Information on rehabilitation was collected from primary caregivers by a physiotherapist during medical assessment camps in the context of BCPR recruitment. To document the type of rehabilitation services received, multiple responses were allowed and, therefore, the numbers presented for this variable are not mutually exclusive. Additionally, the medical records available from primary caregivers were reviewed for any documentation of rehabilitation services. The information has been elaborated in the methodology. (Please see line 120-128 of the unmarked manuscript)

Comment-11: Provide details on how variables added to the analysis were handled.

Our response: Thank you for the query. To maintain the quality of the data, we used a double data entry method. Subsequently, data-entry error checks were performed by running frequencies of all variables to identify any outliers. In case of any missing data or incorrect/suspicious information, the BCPR case record forms (i.e. source documents) were reviewed. If the information was not found in the BCPR form, it was relayed to the field team and where possible the participants were contacted to gather the missing information. Continuous variables were collected as exact values and later recoded and categorised into groups (e.g. age was categorised as follows: 0–4 years, 5–9 years, 10–14 years and 15–18 years). Similarly, the monthly family income data were converted to United States Dollar (US$) (considering 1 US$ = 84.43 Bangladeshi taka (BDT)) and categorised into four family-income groups (i.e. BDT 500–4999 (US$ ⁓6–59), BDT 5000–9999 (US$ ⁓59–118), BDT 10,000–14,999 (US$ ⁓118–178) and BDT 15,000 and above (US$ ⁓178 and above). Comparison between the BCPR cohort and the general population was performed using the 2014 Bangladesh Demographic and Health Survey (BDHS) [National Institute of Population Research and Training (NIPORT), Mitra and Associates and ICF International, 2016] and Household Income and Expenditure Survey [Bangladesh Bureau of Statistics, 2019] data from 2016. The poverty level among the families in the BCPR cohort was estimated using the national poverty lines (at the divisional level) as a cut-off. The proportion of families living below the poverty lines was then compared with the general population of the respective divisions as reported in the Household Income Expenditure Survey (HIES) in Bangladesh [Bangladesh Bureau of Statistics, 2019]. Bivariate analyses were completed to assess the impact of socio-demographic and clinical factors on rehabilitation status. For regression models, ‘Not receiving rehabilitation’ was considered as the main outcome of interest. Factors that were found to be statistically significant in unadjusted logistic regression were fitted into an adjusted logistic regression model. Odds ratios with 95% confidence intervals (CI) were reported. A p value <0.05 was considered significant. All data were analysed using the Statistical Package for the Social Sciences (SPSS) software, version 26 (IBM, Armonk, NY, USA).

The information on data management and analysis has been elaborated in the methodology section. (Please see line 134-158 of the unmarked manuscript)

References:

National Institute of Population Research and Training (NIPORT), Mitra and Associates, ICF International. Bangladesh Demographic and Health Survey 2014. Dhaka, Bangladesh, and Rockville, Maryland, USA: NIPORT, Mitra and Associates, and ICF International: 2016. [cited 2020 Aug]. Available from: https://dhsprogram.com/pubs/pdf/FR311/FR311.pdf

Bangladesh Bureau of Statistics. Preliminary Report on Household Income and Expenditure Survey 2016. Dhaka, Bangladesh: Bangladesh Bureau of Statistics. 2019; [cited 2020 Aug]. Available from: http://bbs.portal.gov.bd/sites/default/files/files/bbs.portal.gov.bd/page/b343a8b4_956b_45ca_872f_4cf9b2f1a6e0/HIES%20Preliminary%20Report%202016.pdf

Comment-12: How was poverty level assessed?

Our response: Thank you for the query. In the BCPR surveillance we collect information related to the household income-expenditure in local currency (i.e. Bangladesh Taka (BDT)) by probing the major food and non-food consumption categories. To estimate the proportion of families living below the poverty line we previously used the international poverty line (US$ 1.90 per day per capita) as a cutoff value. Since we have submitted this manuscript, we have carefully looked into the poverty data. Subsequently, we have identified some additional concern which might worth considering in terms of poverty/income data for our cohort. For instance, the international poverty line US$ 1.90 per day per capita is an absolute poverty line in purchasing power parity (PPP) to allow comparison with other countries [The World Bank, 2019] which is estimated based on the national poverty lines of selected countries. This conversion of the poverty lines is complex and requires consideration of multiple factors as well as careful interpretation [Jolliffe and Prydz, 2016]. Considering the complexity and risk of misinterpretation, we have now excluded this variable (i.e. International poverty line) from our analysis. Instead, we have used the national poverty line (at the divisional level) of Bangladesh to report the proportion of families living below the poverty line. Please see our revised analysis in the Table 1 and the result section (Please see line 178-182 of the unmarked manuscript). We have also added this information in the methodology section of the revised manuscript. (Please see line 147-151 of the unmarked manuscript) 

The national poverty lines are estimated based on household per capita consumption (food and non-food consumption/expenditure) using a detailed questionnaire during the Household Income Expenditure Survey (HIES) in Bangladesh. Though we did not collect that detailed information as part of the BCPR, our methodology is to some extent similar. However, there is a risk of overreporting the poverty level in the BCPR due to the differences in survey tools and depth of information collected compared to the HIES, and several other factors. We have now added this information in the limitation section of our manuscript. (Please see line 370-376 of the unmarked manuscript) 

References:

The World Bank. How is the global poverty line derived? How is it different from national poverty lines? The World Bank; 2020. Available from: https://datahelpdesk.worldbank.org/knowledgebase/articles/193310-how-is-the-global-poverty-line-derived-how-is-it

Jolliffe D, Prydz EB. Estimating international poverty lines from comparable national thresholds. The World Bank; 2016 Mar 17.

Comment-13: 

Results:

• Clarify “Illiterate mothers had 2.1 times (95% CI 1.8-2.5) less chance of receiving rehabilitation services” whether this relates to mothers receiving rehabilitation services of children of mothers with no education.

• Clarify “Children with a monthly income of BDT 5000-9999 (US$ ⁓59-118) were 1.7 [95% CI 1.5-2.0] times less likely to receive rehabilitation services than children with a monthly family income of BDT 15000 (US$ ⁓178) and above”. This is family or household income, not child’s income. Recheck.

Our response: Thank you for the constructive feedback. The sentences have been revised as follows:

‘Children of mothers with no education had 2.1 times (95% CI 1.8-2.5) less chance of receiving rehabilitation services’. (Please see line 224-226 of the unmarked manuscript) ‘Children from families with a monthly income of BDT 5000-9999 (US$ ⁓59-118) were 1.7 [95% CI 1.5-2.0] times less likely to receive rehabilitation services than children with a monthly family income of BDT 15000 (US$ ⁓178) and above’. (Please see line 232-234 of the unmarked manuscript)

Comment-14: Discussion: Nicely written.

Our response: Thank you for the appreciation.

Comment-15: Mention full form at the first mention of an abbreviation

Our response: Thank you for the feedback. We have reviewed the manuscript thoroughly and confirmed to use the abbreviations only after stating the full form at the first mention.

Comment-16: Check format of the paper so that it is in accordance with the journal’s guidelines

Our response: Thank you for the feedback. There were some formatting errors that have been corrected in the revised manuscript. 

Comment-17: Check references, their formatting and citation style as per the journal’s requirements

Our response: Thank you for the feedback. The in-text citations and bibliography have been revised following the journal guidelines.

Comment-18: Check for language edits and sentence structure

Our response: Thank you for the feedback to improve the readability of our manuscript. The manuscript has been reviewed and edited by a professional proofreader.

Yours sincerely,

Gulam Khandaker

On behalf of the study investigators

---

## [Decision Letter · Decision Letter 1]

31 Mar 2021

PONE-D-21-00848R1

Rehabilitation status of children with cerebral palsy in Bangladesh: findings from the Bangladesh Cerebral Palsy Register

PLOS ONE

Dear Dr. Khandaker,

Thank you for submitting your manuscript to PLOS ONE. After careful consideration, we feel that it has merit but does not fully meet PLOS ONE’s publication criteria as it currently stands. Therefore, we invite you to submit a revised version of the manuscript that addresses the points raised during the review process.

Please address comments from one of the reviewers.

We look forward to receiving your revised manuscript.

Kind regards,

Enamul Kabir

Academic Editor

PLOS ONE

Journal Requirements:

Reviewers' comments:

Reviewer's Responses to Questions

**Comments to the Author**

1. If the authors have adequately addressed your comments raised in a previous round of review and you feel that this manuscript is now acceptable for publication, you may indicate that here to bypass the “Comments to the Author” section, enter your conflict of interest statement in the “Confidential to Editor” section, and submit your "Accept" recommendation.

Reviewer #1: (No Response)

Reviewer #2: All comments have been addressed

2. Is the manuscript technically sound, and do the data support the conclusions?

Reviewer #1: Yes

Reviewer #2: Yes

3. Has the statistical analysis been performed appropriately and rigorously? 

Reviewer #1: Yes

Reviewer #2: Yes

4. Have the authors made all data underlying the findings in their manuscript fully available?

Reviewer #1: No

Reviewer #2: Yes

5. Is the manuscript presented in an intelligible fashion and written in standard English?

Reviewer #1: Yes

Reviewer #2: Yes

6. Review Comments to the Author

Reviewer #1: In the methods section, line 96 to line 133 should be written in past tense. Read through the methods section to ensure its written in the correct tenses.

Reviewer #2: Thank you for making the revisions and addressing all comments raised based on the last version of the manuscript. Thanks

7. PLOS authors have the option to publish the peer review history of their article (what does this mean?). If published, this will include your full peer review and any attached files.

Reviewer #1: **Yes: **Dr Tope Olubodun

Reviewer #2: **Yes: **Nukhba Zia

---

## [Author Response · Author response to Decision Letter 1]

5 Apr 2021

05 April 2021

Professor Joerg Heber 

Editor-in-Chief

PLOS ONE

RE: Manuscript ID PONE-D-21-00848

Title: Rehabilitation status of children with cerebral palsy in Bangladesh: findings from the Bangladesh Cerebral Palsy Register

We would like to thank the academic editor and the reviewers for their constructive feedback on our manuscript titled ‘Rehabilitation status of children with cerebral palsy in Bangladesh: findings from the Bangladesh Cerebral Palsy Register’. We have incorporated all of the Academic Editor’s and Reviewers’ feedback into our revised manuscript. Additionally, all statistical methods employed in this study were re-performed and reviewed. The errors and inconsistencies identified during the review have been corrected. Please see below our point-to-point responses to the editor’s and reviewers’ comments and a detailed description of the additional changes that we have made as part of the rebuttal.

Editor:

Comment-1: Please review your reference list to ensure that it is complete and correct. If you have cited papers that have been retracted, please include the rationale for doing so in the manuscript text, or remove these references and replace them with relevant current references. Any changes to the reference list should be mentioned in the rebuttal letter that accompanies your revised manuscript. If you need to cite a retracted article, indicate the article’s retracted status in the References list and also include a citation and full reference for the retraction notice.

Our response: Thank you for the feedback. We have reviewed the reference list. There were some errors in the reference list (e.g. place of publication in reference# 1, 30, 33 and 34; semicolon in reference# 1, 6, 30, 33, 34; erratum in reference#3, 5 and 36; DOI in reference# 4, 8, 9, 20, 24, 27, 38, 41, 42 and 44; publisher name in reference# 40; book chapter format in reference# 11 and 39; author list in reference# 27 and journal name abbreviation in reference# 43) that have been corrected in the revised version of the manuscript. Sincere apologies for the unintentional errors. We confirm that we have not cited any retracted article in this manuscript.

Reviewer: 1

Comment-1: 

In the methods section, line 96 to line 133 should be written in past tense. Read through the methods section to ensure its written in the correct tenses.

Our response: Thank you for the valuable comment. We have revised the methodology section to incorporate the suggestion. Please see lines 96-133.

Additional changes:

As part of this rebuttal, to ensure statistical integrity of the study findings, all statistical procedures were re-performed and reviewed. We subsequently identified that the method of computing dummy variable for age categories (i.e. Age groups 5-9 and 10-14) was incorrect and there was a calculation error in the ‘Number of associated impairments’ variable. As we have corrected these two variables, the unadjusted odds ratios for these variables (Table 3 [row 4 and 5] and Table 4 [row 30-32]) and adjusted odds ratios and corresponding p values for all variables entered into the multiple regression model (Table 5) have also changed to some extent. Considering the corrections, we have now revised the abstract (line 16-23), results (line 260, 267 and 271-277), discussions (line 289, 322-332 and 348-353) and conclusions (line 392) sections of the manuscript (unmarked version) accordingly. Additionally, the lower limit of 95% Confidence Interval of unadjusted odds ratios for Hearing Impairment has been changed from 1.7 to 1.6 (Table 4 [row 27]). We sincerely apologise for these unintentional errors that we have corrected now. However, we would like to confirm that the revised results have not made any major changes to our study findings.

Yours sincerely,

Gulam Khandaker

On behalf of the study investigators

---

## [Editor Report · Decision Letter 2]

12 Apr 2021

Rehabilitation status of children with cerebral palsy in Bangladesh: findings from the Bangladesh Cerebral Palsy Register

PONE-D-21-00848R2

Dear Dr. Khandaker,

We’re pleased to inform you that your manuscript has been judged scientifically suitable for publication and will be formally accepted for publication once it meets all outstanding technical requirements.

Kind regards,

Enamul Kabir

Academic Editor

PLOS ONE
---

## [Editor Report · Acceptance letter]

23 Apr 2021

PONE-D-21-00848R2 

Rehabilitation status of children with cerebral palsy in Bangladesh: findings from the Bangladesh Cerebral Palsy Register 

Dear Dr. Khandaker:

I'm pleased to inform you that your manuscript has been deemed suitable for publication in PLOS ONE. Congratulations! Your manuscript is now with our production department. 

Kind regards, 

on behalf of

Dr. Enamul Kabir 

Academic Editor

PLOS ONE